# Severe infections emerge from commensal bacteria by adaptive evolution

Bernadette C Young[1,2]*, Chieh-Hsi Wu[1], N Claire Gordon[1], Kevin Cole[3], James R Price[3,4], Elian Liu[1,2], Anna E Sheppard[1,5], Sanuki Perera[1,2], Jane Charlesworth[1], Tanya Golubchik[1], Zamin Iqbal[6], Rory Bowden[6], Ruth C Massey[7], John Paul[8,9], Derrick W Crook[1,8,9], Timothy E Peto[1,9], A Sarah Walker[1,9], Martin J Llewelyn[3,4], David H Wyllie[1,10], Daniel J Wilson[1,6,11]*

[1]Nuffield Department of Medicine, Experimental Medicine Division, University of Oxford, Oxford, United Kingdom; [2]Microbiology and Infectious Diseases Department, Oxford University Hospitals NHS Foundation Trust, Oxford, United Kingdom; [3]Department of Infectious Diseases and Microbiology, Royal Sussex County Hospital, Brighton, United Kingdom; [4]Department of Global Health and Infection, Brighton and Sussex Medical School, University of Sussex, Brighton, United Kingdom; [5]NIHR Health Protection Unit in Healthcare Associated Infections and Antimicrobial Resistance at University of Oxford in partnership with Public Health England, Oxford, United Kingdom; [6]Wellcome Trust Centre for Human Genetics, University of Oxford, Oxford, United Kingdom; [7]School of Cellular and Molecular Medicine, University of Bristol, Bristol, United Kingdom; [8]National Infection Service, Public Health England, London, United Kingdom; [9]National Institute for Health Research, Oxford Biomedical Research Centre, Oxford, United Kingdom; [10]Centre for Molecular and Cellular Physiology, Jenner Institute, Oxford, United Kingdom; [11]Institute for Emerging Infections, Oxford Martin School, University of Oxford, Oxford, United Kingdom

*For correspondence:
bernadette.young@ndm.ox.ac.uk (BCY);
daniel.wilson@ndm.ox.ac.uk (DJW)

Competing interests: The authors declare that no competing interests exist.

**Abstract** Bacteria responsible for the greatest global mortality colonize the human microbiota far more frequently than they cause severe infections. Whether mutation and selection among commensal bacteria are associated with infection is unknown. We investigated de novo mutation in 1163 *Staphylococcus aureus* genomes from 105 infected patients with nose colonization. We report that 72% of infections emerged from the nose, with infecting and nose-colonizing bacteria showing parallel adaptive differences. We found 2.8-to-3.6-fold adaptive enrichments of protein-altering variants in genes responding to *rsp*, which regulates surface antigens and toxin production; *agr*, which regulates quorum-sensing, toxin production and abscess formation; and host-derived antimicrobial peptides. Adaptive mutations in pathogenesis-associated genes were 3.1-fold enriched in infecting but not nose-colonizing bacteria. None of these signatures were observed in healthy carriers nor at the species-level, suggesting infection-associated, short-term, within-host selection pressures. Our results show that signatures of spontaneous adaptive evolution are specifically associated with infection, raising new possibilities for diagnosis and treatment.
DOI: https://doi.org/10.7554/eLife.30637.001

## Introduction

Infections remain a leading cause of global mortality, with bacterial pathogens among the greatest concern (**GBD 2015 Mortality and Causes of Death Collaborators, 2016**). However, many of the bacteria imposing the greatest burden of mortality, such as *Staphylococcus aureus*, are frequently

found as commensal components of the body's microbiota (*Turnbaugh et al., 2007*). For them, infection is a relatively uncommon event that is often unnecessary (*Casadevall et al., 2011*; *Méthot and Alizon, 2014*), and perhaps disadvantageous (*Brown et al., 2012*), for onward transmission. Genomics is shedding light on important bacterial traits such as host-specificity, toxicity and antimicrobial resistance (*Sheppard et al., 2013*; *Laabei et al., 2014*; *Chewapreecha et al., 2014*; *Chen et al., 2015*; *Earle et al., 2016*). These approaches offer new opportunities to understand the role of genetics and within-host evolution in the outcome of human interactions with major bacterial pathogens (*Didelot et al., 2016*).

Several lines of evidence support a plausible role for within-host evolution influencing the virulence of bacterial pathogens. Common bacterial infections, including *S. aureus*, are often associated with colonization of the nose by a genetically similar strain. In these patients, the nose is considered the likely source of infection because the nose is more often the site of asymptomatic colonization than any other body site (*von Eiff et al., 2001*; *Kluytmans et al., 1997*; *Yang et al., 2010*). Genome sequencing suggests that bacteria mutate much more quickly than previously accepted, and this confers a potent ability to adapt, for example evolving antimicrobial resistance de novo within individual patients (*Howden et al., 2011*; *Eldholm et al., 2014*). Opportunistic pathogens infecting cystic fibrosis patients have been found to rapidly adapt to the lung environment, with strong evidence of parallel evolution across patients (*Lieberman et al., 2011*; *Marvig et al., 2013*; *Markussen et al., 2014*; *Lieberman et al., 2014*; *Marvig et al., 2015*). However, the selection pressures associated with antimicrobial resistance and opportunistic infections of cystic fibrosis patients may not typify within-host adaptation in common commensal pathogens that have co-evolved with humans for thousands or millions of years (*Moeller et al., 2016*; *Lees et al., 2017*).

Candidate gene studies have demonstrated that substitutions in certain regions, notably quorum-sensing systems such as the *S. aureus* accessory gene regulator (*agr*), arise particularly quickly in vivo and in culture (*Traber et al., 2008*). The *agr* operon encodes a pheromone that coordinates a shift at higher cell densities from production of surface proteins promoting biofilm formation to production of secreted toxins and proteases promoting inflammation and dispersal (*Novick and Geisinger, 2008*). Mutants typically produce the pheromone but no longer respond to it (*Painter et al., 2014*). The evolution of *agr* has been variously ascribed to directional selection (*Sakoulas et al., 2009*), balancing selection (*Robinson et al., 2005*), social cheating (*Pollitt et al., 2014*) and life-history trade-off (*Shopsin et al., 2010*). However, the role of *agr* mutants in infection remains unclear, since they are frequently sampled from both asymptomatic carriage and severe infections (*Smyth et al., 2012*; *Painter et al., 2014*).

Whole-genome sequencing case studies add weight to the idea that within-host evolution plays an important role in infection. In one persistent *S. aureus* infection, a single mutation was sufficient to permanently activate the stringent stress response, reducing growth, colony size and experimentally measured infection severity (*Gao et al., 2010*). In another patient, we found that bloodstream bacteria differed from those initially colonizing the nose by several mutations including loss-of-function of the *rsp* regulator (*Young et al., 2012*). Functional follow-up revealed that the *rsp* mutant expressed reduced cytotoxicity (*Laabei et al., 2015*), but maintained the ability to cause disseminated infection (*Das et al., 2016*). Unexpectedly, we found that bloodstream-infecting bacteria exhibit lower cytotoxicity than nose-colonizing bacteria more generally (*Laabei et al., 2015*). These results raise the question: are unique hallmarks of de novo mutation and selection associated with bacterial evolution in severely infected patients?

We addressed this question by investigating the genetic variants arising from within-patient evolution of *S. aureus* sampled from 105 patients with concurrent nose colonization and blood or deep tissue infection. We annotated variants to test for systematic differences between colonizing and infecting bacteria. We discovered several groups of genes showing significant enrichments of protein-altering variants compared to other genes, indicating adaptive evolution. For genes implicated in pathogenesis, adaptive mutants were limited to infecting bacteria, while other pathways showed adaptation in the nose and infection site. Adaptive enrichments were not observed in asymptomatic carriers, nor between unrelated bacteria, indicating evolution in response to infection-associated, within-host selection pressures. Our results reveal that adaptive evolution of genes involved in regulation, toxin production, abscess formation, cell-cell communication and bacterial-host interaction drives parallel differentiation between commensal constituents of the nose and infecting bacteria, providing new insights into the evolution of infection in a major pathogen.

# Results

## Infecting bacteria are typically descended from the patient's commensal bacteria

We identified 105 patients suffering severe *S. aureus* infections admitted to hospitals in Oxford and Brighton, England, for whom we could recover contemporaneous nose swabs from admission screening. Of the 105 patients, 55 had bloodstream infections, 37 had soft tissue infections and 13 had bone and joint infections (*Table 1*). The infection was most often sampled on the same day as the nose, with an interquartile range of 1 day earlier to 2 days later (*Supplementary file 1*).

To discover de novo mutations within and between *S. aureus* in the nose and infection site, we whole-genome sequenced 1163 bacterial colonies, a median of 5 per site. We detected single-nucleotide polymorphisms (SNPs) and short insertions/deletions (indels) using previously developed, combined reference-based mapping and de novo assembly approaches (*Young et al., 2012*; *Golubchik et al., 2013*; *Iqbal et al., 2012*). We identified 35 distinct strains, defined by multilocus sequence type (ST), across patients (*Supplementary file 1*). As expected (*von Eiff et al., 2001*), most patients possessed extremely closely related nose-colonizing and infecting bacteria, sharing the same ST and differing by 0–66 variants (95 patients). The nose-colonizing and infecting bacteria of nine patients were unrelated, possessing different STs and differing by 9398–50573 variants (e.g. *Figure 1A*). In one further patient, we deemed the nose-colonizing and infecting bacteria to be unrelated despite sharing the same ST because they differed by 1104 variants, far outside the within-ST variation evident in any individual nose or infection site (*Figure 1—figure supplement 1*), and corresponding to around 70 years of divergence based on our previous estimates of within-host evolution (*Young et al., 2012*). In 9/95 patients with extremely closely related nose-colonizing and infecting bacteria, another, unrelated ST was also present in the nose (six patients) or the infection site (three patients); we excluded these unrelated bacteria from further analysis. After excluding variants differentiating unrelated nose-colonizing and infecting bacteria, we catalogued 1322 de novo mutations that we deemed arose within the 105 patients.

In patients with closely related strains, the within-patient population structure was always consistent with a unique migration event from the nose to the infection site, or occasionally, vice versa. Infecting and nose-colonizing bacteria usually formed closely related but distinct populations with no shared genotypes (74/95 patients, e.g. *Figure 1B*), separated by a mean of 5.7 variants. There was never more than one identical genotype between nose-colonizing and infecting bacteria, (21/95 patients, e.g. *Figure 1C*), indicating that the migration event from one population to the other involved a small number of founding bacteria (*Moxon and Murphy, 1978*; *Margolis and Levin, 2007*; *Prajsnar et al., 2012*). In such patients, the shared genotype likely represents the migrating genotype itself. Population structure did not differ significantly between infection types (p=0.38, *Table 1*). Genetic diversity in the nose (mean pairwise distance, $\pi$ = 2.8 variants) was similar to that previously observed in asymptomatic nasal carriers (*Golubchik et al., 2013*) (Reference Panel I, $\pi$ = 4.1, p=0.13), but was significantly lower in the infection site ($\pi$ = 0.6, p=$10^{-10.0}$), revealing limited diversification post-infection.

In most patients, the infection appeared to be descended from the nose. We used 1149 sequences from other patients and carriers (Reference Panel II) to reconstruct the most recent common

**Table 1.** Distribution of infection types and relatedness of nose-colonizing and infecting *S. aureus* among 105 patients revealed by genomic comparison.

| Infection sites | Relation of nose-colonizing to infecting bacteria | | |
|---|---|---|---|
| | Unrelated (≥1104 variants) | Closely related (≤66 variants) | |
| | | Zero shared genotypes | One shared genotype |
| Bloodstream | 4 | 43 | 8 |
| Soft tissue | 4 | 23 | 10 |
| Bone and joint | 2 | 8 | 3 |
| Total | 10 | 74 | 21 |

DOI: https://doi.org/10.7554/eLife.30637.002

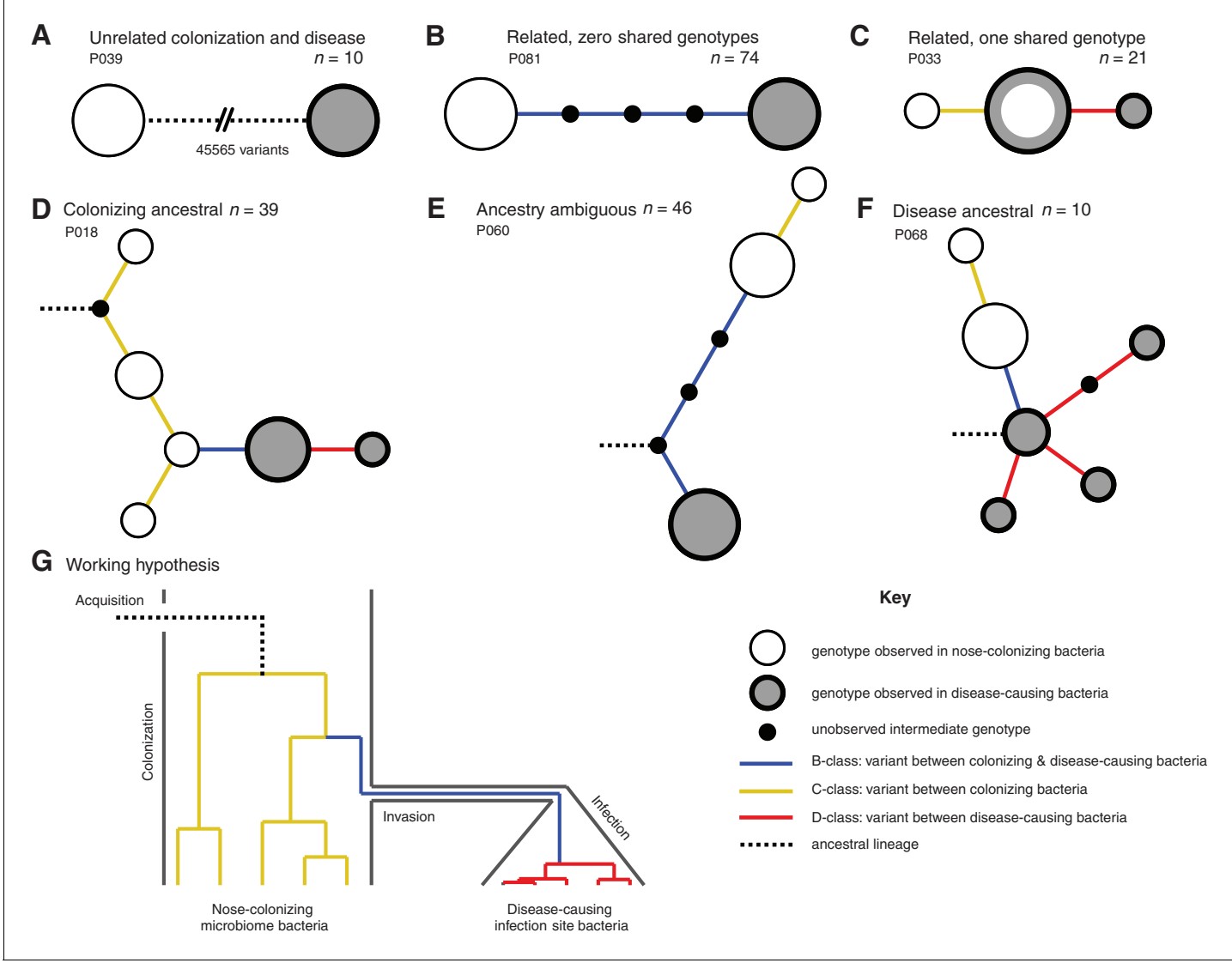

**Figure 1.** Infection-causing *S. aureus* form closely related but distinct populations descended from nose-colonizing bacteria in the majority of infections. Bacteria sampled from the nose and infection site of 105 patients formed one of three population structures, illustrated with example haplotrees: (**A**) Unrelated populations differentiated by many variants. (**B**) Highly related populations separated by few variants. (**C**) Highly related populations with one genotype in common. Reconstructing the ancestral genotype in each patient helped identify the ancestral population: (**D**) Nose-colonizing bacteria ancestral. (**E**) Ambiguous ancestral population. (**F**) Infection site bacteria ancestral. (**G**) Phylogeny illustrating the working hypothesis that variants differentiating highly related nose-colonizing and infection-causing bacteria would be enriched for variants that promote, or are promoted by, infection. In **A–F**, haplotree nodes represent observed genotypes sampled from the nose (white) or infection site (grey), with area proportional to genotype frequency, or unobserved intermediate genotypes (black). Edges represent mutations. Patient identifiers and sample sizes (**n**) are given. In **A–G**, edge color indicates that mutations occurring on those branches correspond to B-class variants between nose-colonizing and infection-causing bacteria (blue), C-class variants among nose-colonizing bacteria (gold) or D-class variants among infection-causing bacteria (red). Black dashed edges indicate ancestral lineages. A B C.

DOI: https://doi.org/10.7554/eLife.30637.003

The following figure supplement is available for figure 1:

**Figure supplement 1.** Distribution of the number of variants identified within 105 severely infected patients, by class.

DOI: https://doi.org/10.7554/eLife.30637.004

ancestor (MRCA) for the 95/105 (90%) patients with related nose-colonizing and infecting bacteria. We thereby distinguished wild type from mutant alleles. In 49 such patients, we could determine the ancestral population. The nose population was likely ancestral in 39/49 (80% of patients with related strains, or 72% of all patients) because all infecting bacteria shared de novo mutations in common that distinguished them from the MRCA, whereas nose-colonizing bacteria did not. In 16 of those, confidence was high because both mutant and ancestral alleles were observed in the nose, confirming it as the origin of the de novo mutation (e.g. *Figure 1D*). Conversely, in 10/49 patients, bacteria colonizing the nose were likely descended from blood or deep tissue infections (20% of patients with related strains, or 18% of all patients) (e.g. *Figure 1F*). Confidence was high for just three of those patients, and they showed unusually high diversity (Supplementary data, P063, P072, P093), suggesting that in persistent infections, infecting bacteria can recolonize the nose.

## Protein-truncating mutants are over-represented within infected patients

To help identify variants that could promote, or be promoted by, infection of the blood and deep tissue by bacteria colonizing the nose, we reconstructed within-patient phylogenies and classified variants by their position in the phylogeny. Sequencing multiple colonies per site enabled us to classify variants into those representing genuine differences *between* nose-colonizing and infection populations (*B*-class), variants specific to the nose-*colonizing* population (*C*-class) and variants specific to the *disease*-causing infection population (*D*-class). We hypothesized that B-class variants would be most enriched for variants promoting, or promoted by, infection, if such variants occur (*Figure 1G*).

We cross-classified variants by their predicted functional effect: synonymous, non-synonymous or truncating within protein-coding sequences, or non-coding (*Table 2*, *Supplementary file 2*). As expected, the prevailing tendency of selection within patients was to conserve protein sequences, with $d_N/d_S$ ratios indicating rates of non-synonymous change 0.55, 0.68 and 0.63 times the rate expected under strict neutral evolution for B-, C- and D-class variants, respectively.

In a longitudinal study of one long-term carrier, we previously reported that a burst of protein-truncating variants punctuated the transition from asymptomatic nose carriage to infection (*Young et al., 2012*). Here, we found a 3.9-fold over-abundance of protein-truncating variants of all phylogenetic classes in infected patients compared to asymptomatic carriers (Reference Panel I, p=0.002, *Table 2*), supporting the conclusion that loss-of-function mutations are disproportionately associated with evolution within infected patients. This may reflect a reduction in the efficiency with which selection removes deleterious protein-truncating mutations, and provides evidence of a systematic difference in selection within severely infected patients.

**Table 2.** Cross-classification of variants within patients by phylogenetic position and predicted functional effect, and comparison to asymptomatic nose carriers.

Neutrality indices (*McDonald and Kreitman, 1991*; *Rand and Kann, 1996*) were defined as the odds ratio of mutation counts relative to synonymous variants in patients versus asymptomatic nose carriers (Reference Panel I). Those significant at p<0.05 and p<0.005 are emboldened and underlined respectively.

| Phylogenetic position | Number of variants (Neutrality index) | | | | |
| | Synonymous | Non-synonymous | Protein truncating | Non-coding | Total |
|---|---|---|---|---|---|
| Patients with severe infections (*n* = 105) | | | | | |
| Between nose-colonization and infection site (B-class) | 93 | 265 (1.1) | **39 (3.1)** | 140 (1.2) | 537 |
| Within nose-colonization (C-class) | 93 | 325 (1.3) | **59 (4.7)** | 145 (1.3) | 622 |
| Within infection site (D-class) | 26 | 82 (1.2) | **15 (4.3)** | 40 (1.3) | 163 |
| Total | 212 | 672 (1.2) | **113 (3.9)** | 325 (1.3) | 1322 |
| Asymptomatic carriers (*Golubchik et al., 2013*) (Reference panel I, for comparison, *n* = 13) | | | | | |
| Within nose-colonization (C-class) | 37 | 97 | 5 | 45 | 184 |

DOI: https://doi.org/10.7554/eLife.30637.005

## Quorum sensing and cell-adhesion proteins exhibit adaptive evolution between nose-colonizing and infecting bacteria

We hypothesized that variants associated with infection would be enriched among the protein-altering B-class variants between the nose and infection site (*Figure 1G*). Therefore, we aggregated mutations by genes in a well-annotated reference genome, MRSA252, and tested each gene for an excess of non-synonymous and protein-truncating B-class variants compared to other genes, taking into account the length of the genes. Aggregating by gene was necessary because 1318/1322 variants were unique to single patients. The two exceptions involved non-coding variants arising in two patients each, one B-class variant 130 bases upstream of *azlC*, an azaleucine resistance protein (SAR0010), and one D-class variant 88 bases upstream of *eapH*1, a secreted serine protease inhibitor (SAR2295) (*Stapels et al., 2014*).

We found a significant excess of five protein-altering B-class variants representing a 58.3-fold enrichment in *agrA*, which encodes the response regulator that mediates activation of the quorum-sensing system at high cell densities (p=$10^{-7.5}$, *Figure 2A*, *Table 3*). The *clfB* gene encoding clumping factor B, which binds human fibrinogen and loricrin (*Foster et al., 2013*), showed an excess of five protein-altering B-class variants, representing a 15.9-fold enrichment that was near genome-wide significance after multiple testing correction (p=$10^{-4.7}$). Both signals of enrichment produced neutrality indices exceeding one, consistent with adaptive evolution (*Supplementary file 3*).

Previously, we identified a truncating mutation in the transcriptional regulator *rsp* to be the most likely candidate for involvement in the progression to infection in one long-term nasal carrier (*Young et al., 2012*). Although we observed just one variant in *rsp* among the 105 patients (3.9-fold enrichment, p=0.27), we found it was a non-synonymous B-class variant resulting in an alanine to proline substitution in the regulator's helix-turn-helix DNA binding domain. In separately published

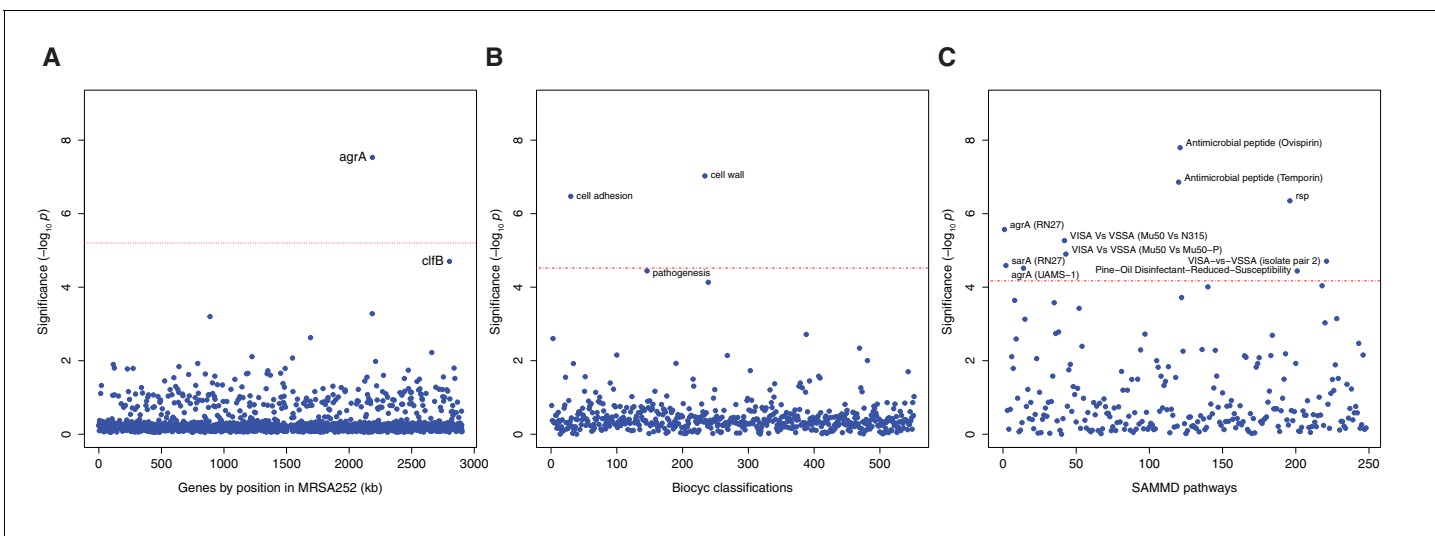

**Figure 2.** Genes, ontologies and pathways enriched for protein-altering substitutions between nose-colonizing and infection-causing bacteria within infected patients. (**A**) Significance of enrichment of 2650 individual genes. (**B**) Significance of enrichment of 552 gene sets defined by BioCyc gene ontologies. (**C**) Significance of enrichment of 248 gene sets defined by SAMMD expression pathways. Genes, pathways and ontologies that approach or exceed a Bonferroni-corrected significance threshold of $\alpha$ = 0.05, weighted for the number of tests per category, (red lines) are named.

DOI: https://doi.org/10.7554/eLife.30637.006

The following figure supplements are available for figure 2:

**Figure supplement 1.** Genes, ontologies and pathways enriched for protein-altering transient variants within nose-colonizing and infection-causing bacteria.

DOI: https://doi.org/10.7554/eLife.30637.007

**Figure supplement 2.** Gene set enrichment analysis of B-class mutants occurring in the nose or the infection site.

DOI: https://doi.org/10.7554/eLife.30637.008

**Figure supplement 3.** Genes, ontologies and pathways enriched for protein-altering variants among longitudinally sampled asymptomatic nasal carriers.

DOI: https://doi.org/10.7554/eLife.30637.009

**Table 3.** Genes, gene ontologies and expression pathways exhibiting the most significant enrichments or depletions of protein-altering B-class variants separating nose and infection site bacteria.

Enrichments below one represent depletions. The total number of variants and genes available for analysis differed by database. A -log₁₀ p-value above 5.2, 4.5 or 4.2 was considered genome-wide significant for loci, gene ontologies or expression pathways respectively (in bold).

| Gene group | No. protein-altering B-class variants | Cumulative lengthof genes (kb) | Enrichment | Significance (-log₁₀ p value) |
|---|---|---|---|---|
| **Locus** | | | | |
| *agrA* | 5 | 0.7 | 58.27 | **7.53** |
| *clfB* | 5 | 2.6 | 15.87 | 4.70 |
| Total | 289 | 2363.8 | | |
| | | | | |
| BioCyc Gene Ontology (*Caspi et al., 2016*) | | | | |
| Cell wall | 18 | 30.9 | 5.02 | **7.03** |
| Cell adhesion | 13 | 17.2 | 6.44 | **6.47** |
| Pathogenesis | 31 | 112.5 | 2.41 | 4.44 |
| Total | 288 | 2359.3 | | |

| SAMMD Expression Pathway | Down-regulated | Up-regulated | Down-regulated | Up-regulated | Down-regulated | Up-regulated | |
|---|---|---|---|---|---|---|---|
| Ovispirin-1 (*Pietiäinen et al., 2009*) | 40 | 7 | 121.2 | 142.9 | 2.65 | 0.39 | **7.80** |
| Temporin L (*Pietiäinen et al., 2009*) | 42 | 14 | 125.1 | 156.1 | 2.78 | 0.74 | **6.86** |
| *rsp* (*Lei et al., 2011*) | 27 | 1 | 61.1 | 13.7 | 3.61 | 0.60 | **6.35** |
| *agrA* (RN27) (*Dunman et al., 2001*) | 9 | 30 | 41.0 | 85.0 | 1.83 | 2.94 | **5.57** |
| VISA-vs-VSSA (Mu50 vs N315) (*Cui et al., 2005*) | 0 | 17 | 0 | 34.4 | 0 | 3.95 | **5.27** |
| VISA-vs-VSSA (Mu50 vs Mu50-P) (*Cui et al., 2005*) | 0 | 17 | 0 | 36.7 | 0 | 3.70 | **4.90** |
| VISA-vs-VSSA (isolate pair 2) (*Howden et al., 2008*) | 14 | 3 | 26.9 | 59.7 | 4.06 | 0.39 | **4.71** |
| *sarA* (RN27) (*Dunman et al., 2001*) | 6 | 23 | 49.9 | 57.7 | 0.97 | 3.22 | **4.59** |
| *agrA* (UAMS-1 OD 1.0) (*Cassat et al., 2006*) | 0 | 5 | 0 | 2.7 | 0 | 14.57 | **4.52** |
| Pine-Oil Disinfectant-Reduced-Susceptibility (*Lamichhane-Khadka et al., 2008*) | 17 | 5 | 36.4 | 23.6 | 3.76 | 1.70 | **4.44** |
| Total | 275 | | 2093.5 | | | | |

DOI: https://doi.org/10.7554/eLife.30637.010

experiments (*Das et al., 2016*), we demonstrated that this and the original mutation induce similar loss-of-function phenotypes which, like *agr* loss-of-function mutants, express reduced cytotoxicity, but maintained an ability to persist, disseminate and cause abscesses in vivo.

We found no significant enrichments of protein-altering variants among D-class variants, but we observed a significant excess of six protein-altering C-class variants in *pbp2* which encodes a penicillin binding protein involved in cell wall synthesis (19.0-fold enrichment, p=10⁻⁶·⁰, *Figure 2—figure supplement 1A*). Pbp2 is an important target of β-lactam antibiotics (*Łeski and Tomasz, 2005*), revealing adaption – potentially in response to antibiotic treatment – in the nose populations of some patients.

## Genes modulated by virulence regulators and antimicrobial peptides show adaptive evolution between colonizing and infecting bacteria

To improve the sensitivity to identify adaptive evolution associated with infection, we developed a gene set enrichment analysis (GSEA) approach in which we tested for enrichments of protein-altering B-class variants among groups of genes. GSEA allowed us to detect signatures of adaptive evolution in groups of related genes that were not apparent when interrogating individual genes.

We grouped genes in two different ways: by gene ontology and by expression pathway. First, we obtained a gene ontology for the reference genome from BioCyc (*Caspi et al., 2016*), which classifies genes into biological processes, cellular components and molecular functions. There were 552 unique gene ontology groupings of two or more genes. We tested for an enrichment among genes belonging to the ontology, compared to the rest of the genes.

Second, we obtained 248 unique expression pathways from the SAMMD database of transcriptional studies (*Nagarajan and Elasri, 2007*). For each expression pathway, genes were classified as up-regulated, down-regulated or not differentially regulated in response to experimentally manipulated growth conditions or expression of a regulatory gene. For each expression pathway, we tested for an enrichment in genes that were up- or down-regulated compared to genes not differentially regulated.

The most significant enrichment for protein-altering B-class variants between nose and infection sites occurred in the group of genes down-regulated by the cationic antimicrobial peptide (CAMP) ovispirin-1 ($p=10^{-7.8}$), with a similar enrichment in genes down-regulated by temporin L exposure ($p=10^{-6.9}$, *Figure 2C*). Like human CAMPs, the animal-derived ovispirin and temporin compounds inhibit epithelial infections by killing phagocytosed bacteria and mediating inflammatory responses (*Pietiäinen et al., 2009*). In response to inhibitory levels of ovispirin and temporin, *agr*, surface-expressed adhesins and secreted toxins are all down-regulated. Collectively, down-regulated genes showed 2.7-fold and 2.8-fold enrichments of adaptive evolution, respectively. Conversely, genes up-regulated in response to CAMPs, including the *vraSR* and *vraDE* cell-wall operons and stress response genes (*Pietiäinen et al., 2009*), exhibited 0.4-fold and 0.7-fold enrichments (i.e. depletions), respectively (*Table 3*). Thus, expression of the genes undergoing adaptive evolution is strongly inhibited in vitro by host-derived antimicrobial peptides.

Genes belonging to the cell wall ontology showed the second most significant enrichment for adaptive evolution ($p=10^{-7.0}$). Genes contributing to this 5.0-fold enrichment included the immuno-globulin-binding *S. aureus* Protein A (*spa*), the serine rich adhesin for platelets (*sasA*), clumping factors A and B (*clfA, clfB*), fibronectin binding protein A (*fnbA*) and bone sialic acid binding protein (*bbp*). The latter four genes contributed to another statistically significant 6.4-fold enrichment of adaptive protein evolution in the cell adhesion ontology ($p=10^{-6.5}$, *Figure 3*). Therefore, there is a general enrichment of surface-expressed host-binding antigens undergoing adaptive evolution.

The *rsp* regulon showed the most significant enrichment among gene sets defined by response to individual bacterial regulators ($p=10^{-6.4}$). Genes down-regulated by *rsp* in exponential phase (*Lei et al., 2011*), including surface antigens and the urease operon, exhibited a 3.6-fold enrichment for adaptive evolution, while up-regulated genes showed 0.6-fold enrichment. So whereas *rsp* loss-of-function mutants were rare per se, genes up-regulated in such mutants were hotspots of within-patient adaptation in infected patients. Since expression is a prerequisite for adaptive protein evolution, this implies there are alternative routes by which genes down-regulated by intact *rsp* can be expressed and thereby play an important role within patients other than direct inactivation of *rsp*.

Loss-of-function in *agr* mutants represent one alternative route, since they exhibit similar phenotypes to *rsp* mutants, with reduced cytotoxicity and increased surface antigen expression, albeit reduced ability to form abscesses (*Das et al., 2016*). We found significant enrichments of genes regulated by *agrA* in two different backgrounds ($p<10^{-4.5}$) and by *sarA* ($p=10^{-4.6}$), underlining the influence of adaptive evolution on both secreted and surface-expressed proteins during infection. We found that expression of genes enriched for protein-altering substitutions was also altered in strains possessing reduced susceptibility to vancomycin, although not in a consistent direction across strains ($p<10^{-4.7}$), and to pine-oil disinfectant ($p=10^{-4.4}$), suggesting such genes may be generally involved in response to harsh environments. All significant signals of enrichment produced neutrality indices exceeding one, consistent with adaptive evolution (*Supplementary file 3*).

Several genes contributed to multiple evolutionary signals, particularly cell-wall anchored proteins involved in adhesion, the infection process and immune evasion (*Foster et al., 2013*), including *fnbA, clfA, clfB, sasA* and *spa*. These multifactorial, partially overlapping signals suggest a large target for selection in adapting to the within-patient environment (*Figure 3*). The fact that we observed no comparable significant enrichments in C-class and D-class protein-altering variants (*Figure 2—figure supplement 1*) indicates that these evolutionary patterns are associated specifically with the infection process.

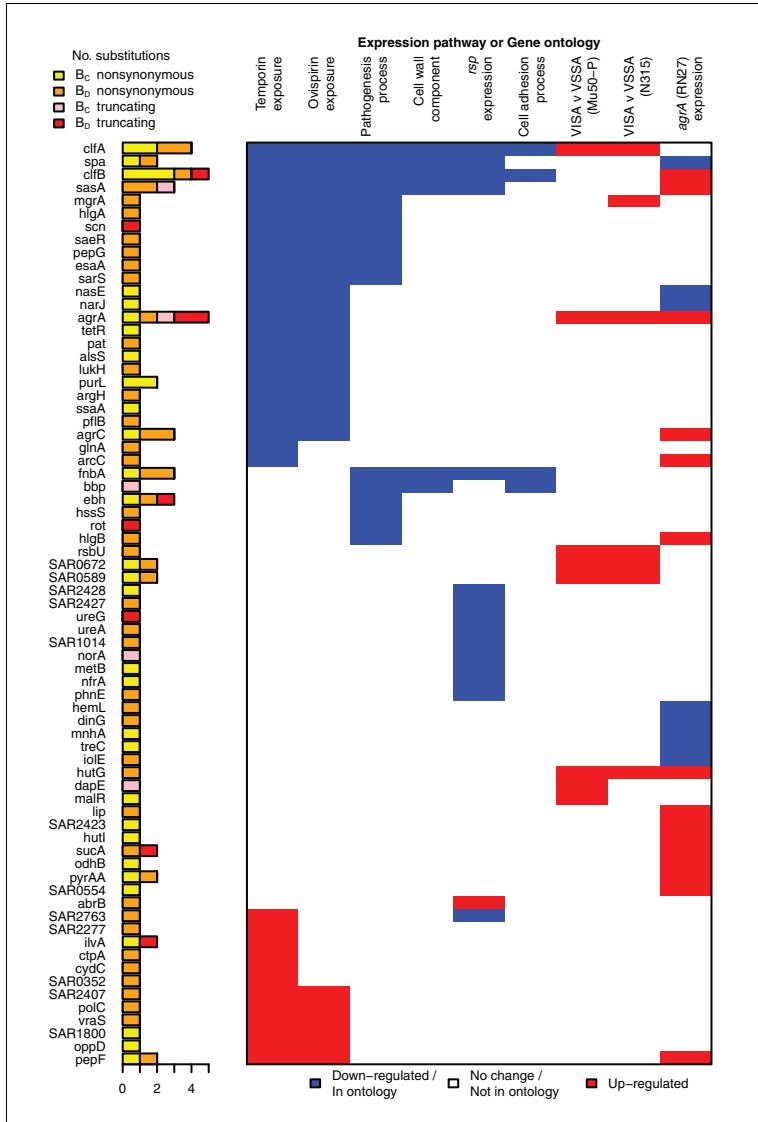

**Figure 3.** All genes contributing to the pathways and ontologies most significantly enriched for protein-altering substitutions between nose-colonizing and infection-causing bacteria. The pathogenesis ontology, in which significant enrichments were observed in infection-causing but not nose-colonizing bacteria, is shown for comparison. Every gene with at least one substitution between nose-colonizing and infection-causing bacteria and which was up- (red) or down- regulated (blue) in one of the pathways or a member of one of the ontologies (blue) is shown. To the left, the number of altering (yellow/orange) and truncating (pink/red) B-class variants is shown, broken down by the population in which the mutant allele was found: nose ($B_C$; yellow/pink) or infection site ($B_D$; orange/red).

DOI: https://doi.org/10.7554/eLife.30637.011

The following figure supplements are available for figure 3:

**Figure supplement 1.** Genes enriched for substitutions between nose-colonizing and infection-causing bacteria within patients are not the most rapidly evolving at the species level.
DOI: https://doi.org/10.7554/eLife.30637.012

**Figure supplement 2.** Gene set enrichment analysis is robust to species-level differences in $d_N/d_S$ between genes.
DOI: https://doi.org/10.7554/eLife.30637.013

## Adaptive evolution in pathogenesis genes is found only in infecting bacteria

Having identified adaptive evolution differentiating nose-colonizing and infection-causing bacteria, we next asked whether the mutant alleles were preferentially found in the nose or infection site. We used 1149 sequences from other patients or carriers (Reference Panel II) to reconstruct the genotype of the MRCA of nose-colonizing and infecting bacteriain each patient, respectively. This allowed us to sub-classify B-class variants by whether the mutant allele was found in the nose-colonizing bacteria ($B_C$-class) or the disease-causing infection site bacteria ($B_D$-class).

A priori, we had expected the enrichments of adaptive evolution to be driven primarily by mutants occurring in the infection-causing bacteria ($B_D$-class). One group of genes showed a signal of such an enrichment among $B_D$-class variants specifically. Genes belonging to the BioCyc pathogenesis ontology were marginally genome-wide significant in $B_D$-class variants, showing a 3.1-fold enrichment ($p=10^{-4.6}$) and a statistically insignificant 1.7-fold enrichment in $B_C$-class variants ($p=0.13$). $B_D$-class mutants driving this differential signal arose in toxins including gamma hemolysin and several regulatory loci implicated in toxicity and virulence regulation: rot, sarS and saeR.

Surprisingly, however, we found that all other significantly enriched gene sets were driven by mutant alleles occurring both in nose-colonizing and infecting bacteria (*Figure 2—figure supplement 2*). This indicates there are common selection pressures in the nose and infection site during the process of infection within patients, leading to convergent evolution across body sites. So while adaptation in pathogenesis genes appears specifically infection-associated, other signals of adaptation in severely infected patients are driven by selection pressures, which might compensate for an altered within-host environment during infection, that are as likely to favor mutants in nose-colonizing bacteria as infecting bacteria.

## Signals of adaptation are specific to infected patients and differ from prevailing signatures of selection

Two lines of evidence show that the newly discovered signatures of within-host adaptive evolution, both in infecting and nose-colonizing bacteria, are unique to evolution in infected patients. To test this theory against the alternative explanation that our approach merely detects the most rapidly evolving proteins, we searched for similar signals in alternative settings: evolution within asymptomatic carriers, and species-level evolution between unrelated bacteria.

There was no significant enrichment of protein-altering variants in any gene, ontology or pathway among 235 variants identified from 10 longitudinally sampled asymptomatic nasal carriers (Reference Panel III, *Figure 2—figure supplement 3*, *Supplementary file 4*). To address the modest sample size, we performed goodness-of-fit tests, focusing on the signals most significantly enriched in patients. We found significant depletions of protein-altering variants in carriers relative to patients in the rsp, agr and sarA regulons ($p=10^{-4.0}$) and the pathogenesis ontology ($p=10^{-3.2}$, *Supplementary file 5*).

Nor were the relative rates of non-synonymous to synonymous substitution ($d_N/d_S$) higher between unrelated *S. aureus* (Reference Panel IV) in the genes that contributed most to the signals associated with adaptation within patients: agrA, agrC clfA, clfB, fnbA and sasA. Although synonymous diversity was somewhat higher than typical in these genes, the $d_N/d_S$ ratios showed no evidence for excess protein-altering change in these compared to other genes (*Figure 3—figure supplement 1*). Accordingly, incorporating this locus-specific variability of $d_N/d_S$ into the GSEA did not affect the results (*Figure 3—figure supplement 2*). Taken together these lines of evidence show that the ontologies, pathways and genes significantly differentiated between nose-colonizing and infecting bacteria arise in response to selection pressures specifically associated with infected patients, and are not repeated in asymptomatic carriers or species-level evolution.

## Discussion

We found that common, life-threatening infections of *S. aureus* are frequently descended from commensal bacteria colonizing the nose. These infections are associated with repeatable patterns of bacterial evolution driven by within-patient mutation and selection. Genes involved in pathogenesis, notably toxins and regulators, showed evidence for adaptation in infecting but not nose-colonizing

bacteria. Surprisingly, other signatures of adaptation occurred in parallel in nose-colonizing and infecting bacteria, affecting genes responding to cationic antimicrobial peptides and the virulence regulators *rsp* and *agr*. Such genes mediate toxin production, abscess formation, immune evasion and bacterial-host binding. Adaptation within both regulator and effector genes reveals that multiple, alternative evolutionary paths are targeted by selection in infected patients.

The signatures of within-patient adaptation that we found differed from prevailing signals of selection at the species level. This discordance means that infection-associated adaptive mutations within patients are rarely transmitted, and argues against a straightforward host-pathogen arms race as the predominant evolutionary force acting within and between patients. Instead, it supports the notion of a life-history trade-off between adaptations favoring colonization and infection distinct from those favoring dissemination and onward transmission (*Shopsin et al., 2010*). Infection of this sort can be characterized in terms of an ecological source-sink model, in which commensal bacteria provide the source for potentially life-threatening infections (*Sokurenko et al., 2006*). Ultimately, the short-term survival advantage gained by infecting bacteria, analogous to the short-term advantage of cancerous cells derived from the host, may lead to the demise of both the host and pathogen, epitomizing a tragedy of the commons (*Rankin et al., 2007*). As such, infection may represent an ever-present risk of mutation among commensal bacteria favored by short-term selection but ultimately incidental or damaging to the bacterial reproductive life cycle.

We did not observe analogous signatures of bacterial adaptation and excess loss-of-function mutations in healthy nose carriers, indicating that risk factors for infections, such as a weakened or over-activated immunological response, comorbidities or medical interventions, may play an important role in creating distinctive selection pressures in infected patients. The effects of such risk factors may be mediated, at least in part, through the selection pressure they exert on commensal bacteria.

The existence of signatures of adaptive substitutions associated with infection raises the possibility of developing new diagnostic techniques and personalizing treatment to the individual patient's commensal bacteria. The ability of genomics to characterize the selective forces driving adaption within the human body in unprecedented detail provides new opportunities to improve experimental models of infection. Ultimately, it may be possible to develop therapies that utilize our new understanding of within-patient evolution to target the root causes of infection from the bacterial perspective.

## Materials and methods

### Patient sample collection

105 patients with severe *S. aureus* infections for whom the organism could be cultured from both admission screening nasal swab and clinical sample were identified prospectively from the microbiological laboratories of hospitals in Oxford and Brighton, England. This study design builds in robustness to potential confounders by matching infection-causing and nose-colonizing bacteria within the same patients. Clinical samples comprised 55 blood cultures and 50 pus, soft tissue, bone or joint samples. The bacteria sampled and sequenced from one patient ('patient S', P005 in this study) have been previously described (*Das et al., 2016*). Five individuals had both blood and another culture-positive clinical sample; we focus analysis on the blood sample. Nasal swabs were incubated in 5% NaCl broth overnight at 37C, then streaked onto SASelect agar (BioRad) and incubated overnight at 37C. We picked five colonies per sample (12 during the pilot phase involving 9 patients), streaked each onto Columbia blood agar and incubated overnight at 37C for DNA extraction. Clinical samples were handled according to the local laboratory standard operating procedure for pus, sterile site and blood cultures. When bacterial growth was confirmed as *S. aureus*, the primary culture plate was retrieved, and multiple colonies were picked. These were streaked onto Columbia blood agar and incubated overnight at 37C for DNA extraction.

### Power calculation for within-patient sample sizes

Sequencing multiple colonies per site allowed us to distinguish genuine genetic substitutions between nose-colonizing and infection-causing bacteria from polymorphic variants restricted to the nose-colonizing or infection-causing bacteria respectively. Following previous studies of within-host

*S. aureus* evolution that we conducted (*Young et al., 2012*; *Golubchik et al., 2013*) and the initial pilot phase in which we sequenced 12 colonies per site, we chose to continue with five colonies per site as a compromise between power to distinguish within-site polymorphisms from true between-site substitutions and the cost of whole genome sequencing. Under a standard neutral model with low mutation rate, five colonies per sample is expected to confer 91% power to correctly distinguish within-site polymorphisms from between-site substitutions, compared to 0% power with one colony per sample. The power calculation is

$$\Pr(0<i<n|0<i) \quad = \frac{\Pr(0<i<n)}{\Pr(0<i)} = \frac{\int_0^1 \Pr(0<i<n)p(f)\mathrm{d}f}{\int_0^1 \Pr(0<i)p(f)\mathrm{d}f}$$
$$= \frac{\int_0^1 (1-f^n-(1-f)^n)f^{-1}\mathrm{d}f}{\int_0^1 (1-(1-f)^n)f^{-1}\mathrm{d}f}$$

where *i* and *f* are the observed sample count and unobserved frequency of a mutant allele in a particular site, *n* is the sample size, and random sampling is assumed. The frequency distribution of a neutral mutant allele, $p(f)$, is derived in (*Sawyer et al., 1992*). If the population were expanding, the power would be greater because mutant alleles would be biased yet more toward low frequencies.

## Reference panels

For comparison to the patient-derived bacteria, we collated previously described samples from other sources to construct four Reference Panels: I. A collection of 131 genomes capturing cross-sectional diversity in the noses of 13 asymptomatic carriers (*Golubchik et al., 2013*), arising from a carriage study of *S. aureus* in Oxfordshire (*Everitt et al., 2014*) (BioProject PRJEB2881). II. A compilation of 95 unrelated samples from the same Oxfordshire carriage study (BioProject accession number PRJEB5225), 145 sequences from a study of within-host evolution of *S. aureus* in three individuals (*Young et al., 2012*) (BioProject PRJEB2862) and 909 sequences from nasal carriage and bloodstream infection used in a study of whole genome sequencing to predict antimicrobial resistance (*Gordon et al., 2014*) (BioProject PRJEB5261). We used these samples to improve our reconstruction of ancestral genotypes in each patient. III. A collection of 237 genomes from longitudinal samples from 10 patients (*Golubchik et al., 2013*; *Gordon et al., 2017*), (BioProject PRJNA380544) arising from the same Oxfordshire carriage study. We used these to compare evolution within patients and asymptomatic carriers. IV. A collection of 16 previously published high-quality closed reference genomes, comprising unrelated isolates mainly of clinical and animal origin: MRSA252 (Genbank accession number BX571856.1), MSSA476 (BX571857.1), COL (CP000046.1), NCTC 8325 (CP000253.1), Mu50 (BA000017.4), N315 (BA000018.3), USA300_FPR3757 (CP000255.1), JH1 (CP000736.1), Newman (AP009351.1), TW20 (FN433596.1), S0385 (AM990992.1), JKD6159 (CP002114.2), RF122 (AJ938182.1), ED133 (CP001996.1), ED98 (CP001781.1), EMRSA15 (HE681097.1) (*Holden et al., 2004*; *Gill et al., 2005*; *Gillaspy, 2006*; *Kuroda et al., 2001*; *Diep et al., 2006*; *Baba et al., 2008*; *Holden et al., 2010*; *Schijffelen et al., 2010*; *Chua et al., 2010*; *Herron-Olson et al., 2007*; *Guinane et al., 2010*; *Lowder et al., 2009*; *Holden et al., 2013*). We used these to contrast species-level evolution to within-patient evolution.

## Whole genome sequencing

For each bacterial colony, DNA was extracted from the subcultured plate using a mechanical lysis step (FastPrep; MPBiomedicals, Santa Ana, CA) followed by a commercial kit (QuickGene; Fujifilm, Tokyo, Japan), and sequenced at the Wellcome Trust Centre for Human Genetics, Oxford on the Illumina (San Diego, CA) HiSeq 2000 platform, with paired-end reads 101 base pairs for nine patients in the pilot phase, and 150 bases in the remainder. We sequenced 62 genomes in duplicate, a technical replication rate of 5.1%; no genetic discordancies were detected within duplicates.

## Variant calling

We used Velvet (*Zerbino and Birney, 2008*) to assemble reads into contigs de novo, and Stampy (*Lunter and Goodson, 2011*) to map reads against two reference genomes: MRSA252 (*Holden et al., 2004*) and a patient-specific reference comprising the contigs assembled for one colony sampled from each patient's nose. Repetitive regions, defined by BLASTing (*Altschul et al., 1990*) the reference genome against itself, were masked prior to variant calling. To obtain multilocus

sequence types (*Enright et al., 2000*), we used BLAST to find the relevant loci, and looked up the nucleotide sequences in the online database at http://saureus.mlst.net/.

Bases called at each position in the reference and those passing previously described (*Young et al., 2012*; *Golubchik et al., 2013*; *Didelot et al., 2012*) quality filters were used to identify single nucleotide polymorphisms (SNPs) from Stampy-based mapping to MRSA252 and the patient-specific reference genomes. We used Cortex (*Iqbal et al., 2012*) to identify SNPs and short indels. Variants found by Cortex were excluded if they had fewer than ten supporting reads or if the base call was heterozygous at more than 5% of reads.

Where physically clustered variants with the same pattern of presence/absence across genomes were found, these were considered likely to represent a single evolutionary event: tandem repeat mutation or recombination. These were de-duplicated to a single variant to avoid inflating evidence of evolutionary events in these regions.

## Variant annotation and phylogenetic classification

Maximum likelihood trees were built to infer bacterial relationships within patients (*Gusfield, 1991*). To prioritize variants for further analysis, they were classified according to their phylogenetic position in the tree: B-class (between nose colonization and infection site), C-class (within nose-colonizing population) and D-class (within infection site population). Variants were cross-classified by their predicted functional effect based on mapping to the reference genome or BLASTing to a reference allele: synonymous, non-synonymous or truncating for protein-coding sequences, or non-coding.

Where variation was found using a patient-specific reference, these variants were annotated by first aligning to MRSA252 using Mauve (*Darling et al., 2004*). If no aligned position in MRSA252 could be found, additional annotated references were used. Where variation was found using Cortex only, the variant was annotated by first locating it by comparing the flanking sequence to MRSA252 and other annotated references using BLAST. MRSA252 orthologs were identified using geneDB (*Logan-Klumpler et al., 2012*) and KEGG (*Kanehisa et al., 2016*).

## Reconstructing ancestral genotypes per patient

We constructed a species-level phylogeny for all bacteria sampled from the 105 patients together with Reference Panel II (unrelated asymptomatic nose-colonization isolates and bloodstream infection isolates) using a two-step neighbor-joining and maximum likelihood approach, based on a whole-genome alignment derived from mapping all genomes to MRSA252. We first clustered individuals into seven groups using neighbour-joining (*Saitou and Nei, 1987*), before resolving the relationships within each cluster by building a maximum likelihood tree using RAxML (*Stamatakis, 2014*), assuming a general time reversible (GTR) model. To overcome a limitation in the presence of divergent sequences whereby RAxML fixes a minimum branch length that may be longer than a single substitution event, we fine-tuned the estimates of branch lengths using ClonalFrameML (*Didelot and Wilson, 2015*). We used these subtrees to identify, for each patient, the most closely related 'nearest neighbor' sampled from another patient or carrier. We employed this nearest neighbor as an outgroup, and used the tree to reconstruct the sequence of the MRCA of colonizing and infecting bacteria for each patient using a maximum likelihood method (*Pupko et al., 2000*) in ClonalFrameML (*Didelot and Wilson, 2015*). This in turn allowed us to identify the ancestral (wild type) and derived (mutant) allele for all variants mapping to MRSA252. For variants not mapping to MRSA252, we repeated the Cortex variant calling analysis, this time including the nearest neighbor, and identified the ancestral allele as the one possessed by the nearest neighbor. This approach allowed us to identify ancestral (wild type) versus derived (mutant) alleles for 97% of within-patient variants. We used the reconstructions of the within-patient MRCA sequences and identity of ancestral vs derived alleles to sub-categorize B-class variants into those in which the mutant allele was found in the nose-colonizing population ($B_C$-class) versus the infection-causing population ($B_D$-class). 521 (97%) of B-class variants were typeable, and in 281 (54%) of these, the mutant allele was found in the infection site population. This allowed us to test for differential enrichments in these two sub-classes.

## Mean pairwise genetic diversity

Separately for the nose site and infection site of each patient, we calculated the mean pairwise diversity $\pi$ as the mean number of variants differing between each pair of genomes. We compared the distributions of $\pi$ between patients and Reference Panel II (13 cross-sectionally sampled asymptomatic nose carriers) using a Mann-Whitney-Wilcoxon test.

## Calculating d$_N$/d$_S$ ratio

For assessing the $d_N/d_S$ ratio within patients, we adjusted the ratio of raw counts of total numbers of non-synonymous and synonymous SNPs by the ratio expected under strict neutrality. We estimated that the rate of non-synonymous mutation was 4.9 times higher than that of synonymous mutation in *S. aureus* based on codon usage in MRSA252 and the observed transition:transversion ratio in non-coding SNPs.

## The neutrality index

To compare the relative $d_N/d_S$ ratios between two groups of variants we computed a Neutrality Index as $R_1/R_2$ where $R_1$ and $R_2$ were the ratio of counts of non-synonymous to synonymous variants in each group respectively (*McDonald and Kreitman, 1991*; *Rand and Kann, 1996*). We compared B-, C- and D-class variants within patients to C-class patients within Reference Panel I (13 cross-sectionally sampled asymptomatic carriers). A Neutrality Index in excess of one indicates a higher $d_N/d_S$ ratio in the former group. We used Fisher's exact test to evaluate the significance of the differences between the groups.

## Gene enrichment analysis

To test for significant enrichment of variants in a particular gene, we employed a Poisson regression in which we modelled the expected numbers of de novo variants across patients in any gene $j$ as $\lambda_0 L_j$ under the null hypothesis of no enrichment, where $\lambda_0$ gives the expected number of variants per kilobase and $L_j$ is the length of gene $j$ in kilobases. We compared this to the alternative hypothesis in which the expected number of variants was $\lambda_i L_i$ for gene $i$, the gene of interest, and $\lambda_1 L_j$ for any other gene $j$. Using R (*R Core Team, 2015*), we estimated the parameters $\lambda_0$, $\lambda_1$ and $\lambda_i$ from the data by maximum likelihood and tested for significance via a likelihood ratio test with one degree of freedom. This procedure assumes no recombination within patients, which was reasonable since we found little evidence of recombination in this study or previously (*Golubchik et al., 2013*), including no within-host genetic incompatibilities, and we removed physically clustered variants associated with possible recombination events. We analyzed all protein-coding genes in MRSA252, testing for an enrichment of variants expected to alter the transcribed protein (both non-synonymous and truncating mutations). These tests were also applied to synonymous mutations and no enrichments were found.

## Gene set enrichment analysis

Since the number of genes outweighed the number of variants detected, we had limited power to detect weak to modest enrichments at the individual gene level. Instead we pooled genes using ontologies from the BioCyc MRSA252 database (*Caspi et al., 2016*) and expression pathways from the SAMMD database of transcriptional studies (*Nagarajan and Elasri, 2007*). The BioCyc database comprises ontologies describing biological processes, cellular components and molecular functions. The SAMMD database groups genes up-regulated, down-regulated or not differentially regulated in response to experimentally manipulated growth conditions or isogenic mutations, usually of a regulatory gene. After excluding ontologies or pathways with two groups, one involving a single gene, and combining ontologies or pathways with identical groupings of genes, we conducted 800 GSEAs in addition to the 2650 ontologies comprised of individual loci. The number of groupings of genes was always two for BioCyc (included/excluded from the ontology) and two or three for SAMMD (up-/down-/un-differentially regulated in the experiment). Again we employed a Poisson regression in which we modelled the expected numbers of variants in any gene $j$ as $\lambda_0 L_j$ under the null hypothesis of no enrichment where $\lambda_0$ gives the expected number of variants per kilobase and $L_j$ is the length of gene $j$ in kilobases. We compared this to the alternative hypothesis in which the expected number of variants was $\lambda_1 L_j$, $\lambda_2 L_j$ or $\lambda_3 L_j$ for gene $j$ depending on the grouping in the ontology/pathway.

Using R, we estimated the parameters $\lambda_0$, $\lambda_1$, $\lambda_2$ and $\lambda_3$ from the data by maximum likelihood and tested for significance via a likelihood ratio test with one or two degrees of freedom, depending on the number of groupings in the ontology/pathway.

### GSEA multiple testing correction

To account for the multiplicity of testing, we adjusted the p-value significance thresholds from a nominal $\alpha$ = 0.05 using the weighted Bonferroni method. We weighted the significance thresholds by the relative number of tests in each category: 2650 genes, 552 BioCyc ontologies and 248 SAMMD expression pathways. This avoids overly stringent multiple testing correction in categories with fewer tests (*Roeder and Wasserman, 2009*), for example, the 248 SAMMD expression pathways, owing to other categories with very large numbers of tests, for example, the 2650 genes. This gave adjusted significance thresholds of $10^{-5.2}$ for genes, $10^{-4.5}$ for BioCyc ontologies and $10^{-4.2}$ for SAMMD expression pathways.

### Longitudinal evolution in asymptomatic carriers

To test whether the patterns of evolution we observed between colonizing and invading bacteria in severely infected patients were typical or unusual, we analyzed Reference Panel III (a collection of 10 longitudinally sampled asymptomatic carriers). Since natural selection is more efficacious over longer periods of time, the longitudinal sampling of these individuals gave us greater opportunity to detect subtle evolutionary patterns in asymptomatic carriers. We characterized variation in these carriers as in the patients. Given the modest sample size and smaller number of variants detected in these individuals (235), we performed GSEA to test for enrichments only in particular genes, ontologies and pathways that were significantly enriched within patients, requiring less stringent multiple testing correction.

### omegaMap analysis

We estimated $d_N/d_S$ ratios between unrelated *S. aureus* to characterize the prevailing patterns of selection at the species level. We used Mauve (*Darling et al., 2004*) to pairwise align 15 reference genomes against MRSA252, that is Reference Panel IV. This allowed us to distinguish orthologs from paralogs in the next step in which we multiply aligned all coding sequences overlapping those in MRSA252 using PAGAN (*Löytynoja et al., 2012*). After removing sequences with premature stop codons, we analyzed each alignment of between two and 16 genes using a modification of omega-Map (*Wilson and McVean, 2006*), assuming all sites were unlinked. We previously showed this assumption, which confers substantial computational efficiency savings, does not adversely affect estimates of selection coefficients (*Wilson et al., 2011*). We estimated variation in $d_N/d_S$ within genes using Monte Carlo Markov chain, running each chain for 10,000 iterations. We assumed exponential prior distributions on the population scaled mutation rate ($\theta$), the transition:transversion ratio ($\kappa$) and the $d_N/d_S$ ratio ($\omega$) with means 0.05, 3 and 0.2, respectively. We assumed equal codon frequencies and a mean of 30 contiguous codons sharing the same $d_N/d_S$ ratio. For each gene, we computed the posterior mean $d_N/d_S$ ratio across sites. This allowed us to rank the relative strength of selection across genes in MRSA252, and to account for differences in $d_N/d_S$, as well as gene length, in the GSEA. We achieved this by modifying the expected number of variants in gene $j$ to be $\lambda_0 \omega_j L_j$ under the null hypothesis of no enrichment versus $\lambda_1 \omega_j L_j$, $\lambda_2 \omega_j L_j$ or $\lambda_3 \omega_j L_j$ under the alternative hypothesis depending on the ontology or pathway, where $\omega_j$ is the posterior mean $d_N/d_S$ in gene $j$.

### Ethical framework

Ethical approval for linking genetic sequences of *S. aureus* isolates to patient data without individual patient consent in Oxford and Brighton in the U.K. was obtained from Berkshire Ethics Committee (10/H0505/83) and the U.K. Health Research Agency [8-05(e)/2010].

## Acknowledgements

We would like to thank Ed Feil, Stephen Leslie, Gil McVean and Richard Moxon for helpful insights and useful discussions. Sequencing reads uploaded to short read archive (SRA) under BioProject PRJNA369475. RNAseq data relating to isolate from P005 (aka 'patient S') previously submitted

under BioProject PRJNA279958. The views expressed in this publication are those of the authors and not necessarily those of the funders. This study was supported by the National Institute for Health Research (NIHR) Oxford Biomedical Research Centre (BRC), a Mérieux Research Grant, the National Institute for Health Research Health Protection Research Unit (NIHR HPRU) in Healthcare Associated Infections and Antimicrobial Resistance at Oxford University in partnership with Public Health England (PHE) (grant HPRU-2012–10041), and the Health Innovation Challenge Fund (a parallel funding partnership between the Wellcome Trust (grant WT098615/Z/12/Z) and the Department of Health (grant HICF-T5-358)). TEP and DWC are NIHR Senior Investigators. DJW and ZI are Sir Henry Dale Fellows, jointly funded by the Wellcome Trust and the Royal Society (Grants 101237/Z/13/Z and 102541/Z/13/Z). BCY is a Research Training Fellow funded by the Wellcome Trust (Grant 101611/Z/13/Z). We acknowledge the support of Wellcome Trust Centre for Human Genetics core funding (Grant 090532/Z/09/Z).

## Additional information

### Funding

| Funder | Grant reference number | Author |
|---|---|---|
| Wellcome | Health Innovation Challenge Fund WT098615/Z/12/Z | Derrick W Crook |
| National Institute for Health Research | Oxford NIHR Biomedical Research Centre | Derrick W Crook<br>Timothy E Peto |
| Department of Health | Health Innovation Challenge Fund HICF-T5-358 | Derrick W Crook |
| Royal Society | Sir Henry Dale Fellowship 101237/Z/13/Z | Daniel J Wilson |
| Institut Mérieux | Mérieux Research Grant | Rory Bowden<br>Derrick W Crook<br>Daniel J Wilson |
| Public Health England | HPRU in Healthcare Associated Infections and Antimicrobial Resistance HPRU-2012-10041 | Derrick W Crook<br>Timothy E Peto<br>A Sarah Walker |
| National Institute for Health Research | HPRU in Healthcare Associated Infections and Antimicrobial Resistance HPRU-2012-10041 | Derrick W Crook<br>Timothy E Peto<br>A Sarah Walker |
| Wellcome | Sir Henry Dale Fellowship 101237/Z/13/Z | Daniel J Wilson |
| Wellcome | Sir Henry Dale Fellowship 102541/Z/13/Z | Zamin Iqbal |
| Royal Society | Sir Henry Dale Fellowship 102541/Z/13/Z | Zamin Iqbal |
| Wellcome | Research Training Fellowship 101611/Z/13/Z | Bernadette C Young |
| Wellcome | Wellcome Trust Centre for Human Genetics core funding 090532/Z/09/Z | Rory Bowden |

The funders had no role in study design, data collection and interpretation, or the decision to submit the work for publication.

### Author contributions

Bernadette C Young, Study design, Sample collection, DNA extraction, Bioinformatics, Analysis, Writing; Chieh-Hsi Wu, Bioinformatics, Analysis, Writing; N Claire Gordon, Sample collection, DNA

extraction; Kevin Cole, Elian Liu, Sanuki Perera, DNA extraction; James R Price, Sample collection; Anna E Sheppard, Jane Charlesworth, Tanya Golubchik, Zamin Iqbal, Bioinformatics; Rory Bowden, Ruth C Massey, Study design, Interpretation; John Paul, Derrick W Crook, Timothy E Peto, A Sarah Walker, Martin J Llewelyn, Study design, Sample collection, Interpretation; David H Wyllie, Study design, Analysis; Daniel J Wilson, Study design, Analysis, Writing

### Author ORCIDs
Bernadette C Young (ID) https://orcid.org/0000-0001-6071-6770
Ruth C Massey (ID) https://orcid.org/0000-0002-8154-4039
Daniel J Wilson (ID) https://orcid.org/0000-0002-0940-3311

### Ethics
Human subjects: Ethical approval for linking genetic sequences of S. aureus isolates to patient data without individual patient consent in Oxford and Brighton in the U.K. was obtained from Berkshire Ethics Committee (10/H0505/83) and the U.K. Health Research Agency [8-05(e)/2010].

### Decision letter and Author response
Decision letter https://doi.org/10.7554/eLife.30637.065
Author response https://doi.org/10.7554/eLife.30637.066

## Additional files

### Supplementary files
• Supplementary file 1. List of all cultures included in the site, the site of infection (and any known source if bloodstream), number of isolates sequenced from each site, ST or CC by in silico MLST, number of variants found at each site and the mean pair-wise difference comparing isolates.
DOI: https://doi.org/10.7554/eLife.30637.014

• Supplementary file 2. List of all variants found within patients with *S. aureus* infections, location on shared reference (MRSA252), or position and reference genome name and accession number if variant could not be localized on MRSA252. Each variant is described by the alleles found, its location in gene, the predicted effect on gene product and the location of the variant on the phylogenetic tree.
DOI: https://doi.org/10.7554/eLife.30637.015

• Supplementary file 3. Neutrality indices show signals of adaptation among the genes, gene ontologies and expression pathways most significantly enriched for protein-altering B-class variants. Neutrality indices (NIs, *41,42*) were calculated as the odds ratio of the number of protein-altering to synonymous variants among B-class versus C/D-class variants. These tests are less powerful than the Poisson regression likelihood ratio tests used to detect gene or gene set enrichment of protein-altering B-class variants (*Table 3*); we present them to demonstrate that the direction of enrichment was consistent with adaptation (NI > 1). To mitigate the reduced power, we calculated the expected numbers of protein-altering B-class variants from the numbers of protein-altering C/D-class variants, synonymous B-class variants and synonymous C/D-class variants by pooling them across all genes. This was justified by the absence of evidence for within-patient recombination and lack of enrichment signals among synonymous variants and C/D class protein-altering variants. A one-tailed Poisson test in R (*R Core Team, 2015*) was used to test NI > 1 (significant NIs at $p<0.05$ in bold).
DOI: https://doi.org/10.7554/eLife.30637.016

• Supplementary file 4. List of all variants found within long term asymptomatic carriers, location on shared reference (MRSA252), or position and reference genome name and accession number if variant was not localized on MRSA252. Each variant is described by the alleles found, its location in gene and the predicted effect on gene product.
DOI: https://doi.org/10.7554/eLife.30637.017

• Supplementary file 5. For all ontologies showing enrichment in within-patient $B_D$-class variants, we identified the genes with variants contributing to the signal. We counted the number of protein-altering variants in these genes within patients, and compared to the number in long-term asymptomatic carriers. p-Values calculated using Fisher's exact test. *Variant totals are different for

SAMMD pathways (*rsp, agrA, sarA*) and BioCyc ontologies (cell wall, cell adhesion, pathogenesis) because pathway information is available for a different number of loci in each database.
DOI: https://doi.org/10.7554/eLife.30637.018

• Transparent reporting form
DOI: https://doi.org/10.7554/eLife.30637.019

## Major datasets

The following dataset was generated:

| Author(s) | Year | Dataset title | Dataset URL | Database, license, and accessibility information |
|---|---|---|---|---|
| Bernadette C Young, Chieh-Hsi Wu, N Claire Gordon, James R Price, Kevin Cole, Elian Liu, Anna E Sheppard, Sanuki Perera, Tanya Golubchik, Zamin Iqbal, Rory Bowden, Ruth C Massey, John Paul, Derrick W Crook, Timothy E Peto, A Sarah Walker, Martin J Llewelyn, David H Wyllie, Daniel J Wilson | 2017 | Illumina Sequencing Data | https://www.ncbi.nlm.nih.gov/bioproject/?term=PRJNA369475 | Publicly available at NCBI BioProject (accession no. PRJNA369475) |

The following previously published datasets were used:

| Author(s) | Year | Dataset title | Dataset URL | Database, license, and accessibility information |
|---|---|---|---|---|
| Tanya Golubchik, Elizabeth M. Batty, Ruth R. Miller, Helen Farr, Bernadette C. Young, Hanna Larner-Svensson, Rowena Fung, Heather Godwin, Kyle Knox, Antonina Votintseva, Richard G. Everitt, Teresa Street, Madeleine Cule, Camilla L. C. Ip, Xavier Didelot, Timothy E. A. Peto, Rosalind M. Harding, Daniel J. Wilson, Derrick W. Crook, Rory Bowden | 2013 | Reference Panel I | https://www.ncbi.nlm.nih.gov/bioproject/PRJEB2881 | Publicly available at NCBI BioProject (accession no. PRJEB2881) |

| | | | | |
|---|---|---|---|---|
| Richard G. Everitt, Xavier Didelot, Elizabeth M. Batty, Ruth R Miller, Kyle Knox, Bernadette C. Young, Rory Bowden, Adam Auton, Antonina Votintseva, Hanna Larner-Svensson, Jane Charlesworth, Tanya Golubchik, Camilla L. C. Ip, Heather Godwin, Rowena Fung, Tim E. A. Peto, A. Sarah Walker, Derrick W. Crook & Daniel J. Wilson | 2014 | Reference Panel IIa | https://www.ncbi.nlm.nih.gov/bioproject/PRJEB5225 | Publicly available at NCBI BioProject (accession no. PRJEB5225) |
| Bernadette C. Young, Tanya Golubchik, Elizabeth Batty, Rowena Fung, Hanna Larner-Svensson, Antonina Votintseva, Ruth Miller, Heather Goodwin, Kyle Knox, Richared Everitt, Zamin Iqbal, Andrew Rimer, Madeline Cule, Camilla Ip, Xavier Didelot, Rosalind Harding, Peter Donnelly, Timothy E Peto, Derrick W Crook, Rory Bowden, Daniel J Wilson | 2012 | Reference Panel IIb | https://www.ncbi.nlm.nih.gov/bioproject/PRJEB2862 | Publicly available at NCBI BioProject (accession no. PRJEB2862) |
| N Claire Gordon, JR Price, K Cole, R Everitt, M Morgan, J Finney, AM Kearns, B Pichon, BC Young, DJ Wilson, MJ Llewelyn, J. Paul, TEA. Peto, DW Crooa, AS Walker, T Golubchik | 2014 | Reference Panel IIc | https://www.ncbi.nlm.nih.gov/bioproject/PRJEB5261 | Publicly available at NCBI BioProject (accession no. PRJEB5261) |
| N Claire Gordon, B Pichon, T Golubchik, DJ Wilson, John Paul, DS Blanc, Kevin Cole, J Collins, N Cortes, M Cubbon, FK Gould, PJ Jenks, M Llewelyn, JQ Nash, JM Orendi, K Paranthaman, J Price, L Senn, HL Thomas, S Wyllie, DW Crook, Timothy Peto, AS Walker, AM Kearns | 2017 | Reference Panel III | https://www.ncbi.nlm.nih.gov/bioproject/PRJNA380544 | Publicly available at NCBI BioProject (accession no. PRJNA380544) |
| Matthew Holden | 2004 | Reference Panel IV MRSA252 | https://www.ncbi.nlm.nih.gov/nuccore/BX571856.1 | Publicly available at NCBI Nucleotide (accession no. BX571856.1) |

| | | | | |
|---|---|---|---|---|
| Matthew Holden | 2004 | Reference Panel IV MSSA476 | https://www.ncbi.nlm.nih.gov/nuccore/BX571857.1 | Publicly available at NCBI Nucleotide (accession no. BX571857.1) |
| SR Gill | 2005 | Reference Panel IV COL | https://www.ncbi.nlm.nih.gov/nuccore/CP000046.1 | Publicly available at NCBI Nucleotide (accession no. CP0000 46.1) |
| AF Gillaspy | 2006 | Reference Panel IV NCTC 8325 | https://www.ncbi.nlm.nih.gov/nuccore/CP000253.1 | Publicly available at NCBI Nucleotide (accession no. CP000 253.1) |
| M Kuroda | 2001 | Reference Panel IV Mu50 | https://www.ncbi.nlm.nih.gov/nuccore/BA000017.4 | Publicly available at NCBI Nucleotide (accession no. BA000017.4) |
| M Kuroda | 2001 | Reference Panel IV N315 | https://www.ncbi.nlm.nih.gov/nuccore/BA000018.3 | Publicly available at NCBI Nucleotide (accession no. BA000018.3) |
| BA Diep | 2006 | Reference Panel IV USA300_FPR3757 | https://www.ncbi.nlm.nih.gov/nuccore/CP000255.1 | Publicly available at NCBI Nucleotide (accession no. CP000 255.1) |
| A Copeland | 2007 | Reference Panel IV JH1 | https://www.ncbi.nlm.nih.gov/nuccore/CP000736.1 | Publicly available at NCBI Nucleotide (accession no. CP000 736.1) |
| T Baba | 2008 | Reference Panel IV Newman | https://www.ncbi.nlm.nih.gov/nuccore/AP009351.1 | Publicly available at NCBI Nucleotide (accession no. AP00 9351.1) |
| Matthew Holden | 2010 | Reference Panel IV TW20 | https://www.ncbi.nlm.nih.gov/nuccore/FN433596.1 | Publicly available at NCBI Nucleotide (accession no. FN433596.1) |
| MJ Schijffelen | 2010 | Reference Panel IV S0385 | https://www.ncbi.nlm.nih.gov/nuccore/AM990992.1 | Publicly available at NCBI Nucleotide (accession no. AM990 992.1) |
| K Chua | 2010 | Reference Panel IV JKD6159 | https://www.ncbi.nlm.nih.gov/nuccore/CP002114.2 | Publicly available at NCBI Nucleotide (accession no. CP00 2114.2) |
| Herron-Olson | 2007 | Reference Panel IV RF122 | https://www.ncbi.nlm.nih.gov/nuccore/AJ938182.1 | Publicly available at NCBI Nucleotide (accession no. AJ938182.1) |
| CM Guinane | 2010 | Reference Panel IV ED133 | https://www.ncbi.nlm.nih.gov/nuccore/CP001996.1 | Publicly available at NCBI Nucleotide (accession no. CP00 1996.1) |
| BV Lowder | 2009 | Reference Panel IV ED98 | https://www.ncbi.nlm.nih.gov/nuccore/CP001781.1 | Publicly available at NCBI Nucleotide (accession no. CP00 1781.1) |
| Matthew Holden | 2013 | Reference Panel IV EMRSA15 | https://www.ncbi.nlm.nih.gov/nuccore/HE681097.1 | Publicly available at NCBI Nucleotide (accession no. HE681097.1) |

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
