## [Decision Letter]

Thank you for submitting your article "Severe infections emerge from commensal bacteria by adaptive evolution" for consideration by *eLife*. Your article has been reviewed by three peer reviewers, and the evaluation has been overseen by a Reviewing Editor and Gisela Storz as the Senior Editor. The reviewers have opted to remain anonymous.

The reviewers have discussed the reviews with one another and the Reviewing Editor has drafted this decision to help you prepare a revised submission.

Summary:

The reviewers find that the paper describes an interesting study of a large data set of *Staphylococcus aureus* genomes representing isolates from carriage and disease in the same individuals. By analysing genetic changes distinguishing the carriage and disease isolates, the study sheds light on the within-host adaptive evolution in an important opportunistic pathogen.

1) Using genome-level resolution, and taking into account within-host diversity, the study demonstrates that the invasive disease is primarily caused by the carried strain.

2) There were repeatable patterns of evolution in the infection isolates indicative of within patient mutation and selection and these differed from prevailing signals of selection at the species level.

3) Adaptations during infection are not necessarily advantageous for transmission, and therefore may disrupt the reproductive life cycle of *S. aureus*.

4) Genetic diversity was much lower amongst infecting isolates compared to colonizing isolates, a pattern which suggests a bottleneck or small founding population for infection.

5) There is an excess of protein altering variation in the variants that differentiate carriage isolates from disease isolates, but not in those found between carriage isolates or between disease isolates.

6) These variants are shown to be concentrated in genes and pathways relevant to host-interaction and disease causation, are not subject to excess variation in the general population.

The analysis is both detailed and comprehensive and uses appropriate tools and techniques, backed up with rigorous statistical tests. The study is based on a large cohort of patients, and includes multiple isolates from each condition (carriage and disease) in each patient, and provides and unprecedented view of the population structures in disease vs. colonization.

Essential revisions:

1) In referring to the origin of the *S. aureus* within the host, the authors use a mixture of terms. For example, the title refers to 'commensal bacteria', in the abstract there is refer to infections emerging from the 'nose microbiota', and also 'nose-colonizing bacteria'. In the introduction there is mention of 'asymptomatic carriage' and 'commensal components of the body's microbiota' and in the Discussion section there is 'commensal bacteria colonizing the nose'. Given the journal's general audience, the manuscript would benefit from some more general information in the introduction about *S. aureus* relationship with the host and carriage, i.e. the observation about variable carriage in the human population (for example Kluytmans, et al., 1997), and that the nose is regarded as the primary colonization site, although there are other colonization sites in humans (see Yang et al., 2010; see following points).

2) The authors could also aid interpretation by being more consistent with the use of terms to describe origin of the *S. aureus* within the host, and also use them to accurately represent the sampling that has been conducted. For example, the authors have sequenced colonies that have been selectively grown from nose swabs, and therefore it is confusing to refer to nose microbiota. Experimentally, the microbiota of the patients has not been genomically characterized, therefore it is somewhat misleading to refer to the microbiota in this way. Additionally, it infers that *S. aureus* is a constituent part of the microbiota, which is not the case (see point above Kluytmans et al., 1997).

3) *S. aureus* can colonize multiple sites-the pharynx, groin, axilla, rectum, etc. (Yang et al., 2010). In the description of the results there appears to be an assumption that the nares population represents the entire colonizing *S. aureus* population. The experimental design means that additional colonizing *S. aureus* diversity elsewhere on these individuals will not be captured, therefore the interpretation of the patterns of diversity they observe needs to be considered and qualified more fully in the discussions.

4) The results suggest that it is not necessarily the case that the infecting population comes from the host, as not every *S. aureus* infected patient has nares *S. aureus* colonization, and also their data indicate that ~10% of their isolates are unrelated. Looking at Supplementary file 1, it also seems that at least a few of the infecting populations were clearly mixed with distinct STs or outliers (patients 23, 61, and 90), as were a few of the colonizing populations (patients 29, 33, 36, 45, 77, 94). This complicates interpretation further, as it suggests that the infecting populations did not necessarily evolve on or within the patient, but may have been at least in part acquired independently. Given both of these issues, interpreting the direction of infection and colonization (as in Figure 1) seems problematic. This would then seem to call into question the classification of variants into the B, C, D classes. Can the authors address these concerns to strengthen their classification?

5) The authors refer to some of the changes that they see in disease isolates as "adaptive" changes, and define adaptive changes in the Introduction as "evolution in response to disease-associated, within-host selection pressures." However, the definition they seem to use throughout the text is more operational: (Introduction) "enrichments in protein altering variants". A clearer definition is warranted. Can the authors clarify if they mean enrichments above baseline or expected rates as might be seen in "relaxed" purifying selection or neutrality, or are they referring more strictly to situations in which *d_N_/d_S_* values are significantly greater than 1?

6) An important distinction in the definition of adaptation might be between adaptation that happened at the colonizing site that was later co-opted in disease, and evolution that truly was "disease-associated". It may be interesting to consider Gould and Vrba's idea of "exaptation" in this context, where changes that evolved in another selective context become beneficial, for another reason, in a new context. It seems possible that the variants that arise in invasive disease may have evolved much earlier under selection in the colonizing site. In this regard, it may be possible to estimate the timing of certain mutations using molecular clock approaches (especially given the multiple isolates collected from each individual). Even if the mutations in question did, in fact, arise very quickly during the process of invasion, it would be interesting to know how much these rates differ from expected rates of evolution. Adding an analysis of inferred rates of evolution would greatly enhance this manuscript.

7) Toxicity is used throughout the text and is a general term, which can encompass multiple proteins and potential effects. It does not specify the kind of toxicity (e.g. cell death, superantigen activity), target of the toxicity, the effector of the toxicity, or the way it is measured. The general concept of "toxicity" is qualitatively different from phenotypes like "abscess formation" or "quorum sensing" which are specific and measureable. The authors could add clarity using a term like "toxin production", or specifying, "in vitro cell death".

---

## [Author Response]

Essential revisions:1) In referring to the origin of the S. aureus within the host, the authors use a mixture of terms. For example, the title refers to 'commensal bacteria', in the abstract there is refer to infections emerging from the 'nose microbiota', and also 'nose-colonizing bacteria'. In the introduction there is mention of 'asymptomatic carriage' and 'commensal components of the body's microbiota' and in the Discussion section there is 'commensal bacteria colonizing the nose'. Given the journal's general audience, the manuscript would benefit from some more general information in the introduction about S. aureus relationship with the host and carriage, i.e. the observation about variable carriage in the human population (for example Kluytmans, et al., 1997), and that the nose is regarded as the primary colonization site, although there are other colonization sites in humans (see Yang et al., 2010; see following points).

Thank you for highlighting this important issue. From the outset, we have tried to assist comprehension for a general audience by avoiding field-specific terms such as ‘isolates’ and we subsequently identified mixed terminology as a difficulty in our drafting process, so I am grateful to the reviewers for flagging that our efforts to clarify the usage have not gone far enough. The paper now clarifies throughout that colonizing bacteria refers to the nose. I have adopted rigid use of infection in preference to disease and expurgated invasion as an unnecessary qualifier. I have cited the suggested papers and expanded on the points immediately relevant to this study in the Introduction.

2) The authors could also aid interpretation by being more consistent with the use of terms to describe origin of the S. aureus within the host, and also use them to accurately represent the sampling that has been conducted. For example, the authors have sequenced colonies that have been selectively grown from nose swabs, and therefore it is confusing to refer to nose microbiota. Experimentally, the microbiota of the patients has not been genomically characterized, therefore it is somewhat misleading to refer to the microbiota in this way.

I have removed mention of the microbiota extensively, in particular wherever it could mislead readers into thinking we have performed metagenomic sequencing, rather than selectively growing *S. aureus*.

Additionally, it infers that S. aureus is a constituent part of the microbiota, which is not the case (see point above Kluytmans et al., 1997)

I regret that I do not follow this point. From the bacterium’s perspective, the human nose (and other body sites) is its natural niche, so surely it is accurate to define *S. aureus* as a constituent part of the nose microbiota, present in around 30% of healthy adults at any one time.

3) S. aureus can colonize multiple sites-the pharynx, groin, axilla, rectum, etc. (Yang et al., 2010). In the description of the results there appears to be an assumption that the nares population represents the entire colonizing S. aureus population. The experimental design means that additional colonizing S. aureus diversity elsewhere on these individuals will not be captured, therefore the interpretation of the patterns of diversity they observe needs to be considered and qualified more fully in the discussions.

It is true that *S. aureus* colonize body sites other than the nose less frequently and, as requested, in the Introduction we now make this plain, citing the additional reference.

I certainly wouldn’t want our claims to rest on a false assumption that the nose represented the entire colonizing population. I don’t believe there are any points in the paper where this occurs, I believe that our interpretations have always carefully considered this caveat, and I do not believe there are any conclusions in the Discussion section or elsewhere that rest on such an assumption. The improvements in terminology, where reference to colonization is now always qualified as nose-colonization, should make this clear when perhaps previously it wasn’t.

4) The results suggest that it is not necessarily the case that the infecting population comes from the host, as not every S. aureus infected patient has nares S. aureus colonization, and also their data indicate that ~10% of their isolates are unrelated. Looking at Supplementary file 1, it also seems that at least a few of the infecting populations were clearly mixed with distinct STs or outliers (patients 23, 61, and 90), as were a few of the colonizing populations (patients 29, 33, 36, 45, 77, 94). This complicates interpretation further, as it suggests that the infecting populations did not necessarily evolve on or within the patient, but may have been at least in part acquired independently. Given both of these issues, interpreting the direction of infection and colonization (as in Figure 1) seems problematic. This would then seem to call into question the classification of variants into the B, C, D classes. Can the authors address these concerns to strengthen their classification?

The reviewers are absolutely right that the valid interpretation of the results of any within-host study relies on the identification of variants that truly arose within-host. This is a point on which we have taken great care:

As the reviewers note, we began the Results section by stating clearly that in 10 patients, the invasive *S. aureus* were unrelated to the nose-colonizing *S. aureus*. At the end of this paragraph, we stated that we excluded variants differentiating unrelated STs from further analysis. I have now clarified that in one of these 10 patients, nose-colonizing and infecting bacteria possessed the same ST but differed by 1104 variants, far outside the within-ST variation evident in any individual nose or infection site (Figure 1—figure supplement 1), and corresponding to around 70 years of divergence based on our previous estimates of within-host evolution (Young et al., 2012), which are consistent with others’ estimates from between-host evolution (e.g. Harris et al., 2010). The other nine with unrelated STs differed by 9398-50573 variants.

The key consideration is what threshold to apply to distinguish related from unrelated nose and infecting bacteria. Figure 1—figure supplement 1 is designed to make transparent our reasoning: as the legend states, “When the number of B-class variants was 66 or less, nose-colonizing and infecting bacteria were considered related, since a similar range of (C-class) diversity was observed within the nose-colonizing populations of bacteria with the same multilocus sequence type. When the number of B-class variants was 1104 or more, nose-colonizing and infecting bacteria were considered unrelated.”

The choice of threshold was clear-cut in these patients, as can be seen from comparing the empirical distributions of B, C and D-class variants in Figure 1—figure supplement 1. This is one of the benefits of having sequenced multiple colonies per body site. However, as noted above we were also guided by past studies that we (and others) conducted, which show the *S. aureus* substitution rate is around 8 variants per genome per year. So the 66-variant threshold represents 4.1 years since the nose and infection populations diverged, within the range of diversity we saw among C-class variants.

The reviewers are right to note that there were interesting signals in our data that we did not dwell on because the focus of the study was the role of de novo mutation and selection within patients. For example, we did indeed show in Supplementary file 1 that in nine of the 95 patients with extremely closely related nose-colonizing and infecting bacteria, there were also present *S. aureus* of another, unrelated ST. We now specifically alert readers at the beginning of the Results, making clear that these unrelated populations were excluded from further analysis. It would be tempting to devote more space to the multiply infected patients but the paper is already complex and a sample size of three is too small to draw meaningful conclusions without the ethical approval to explore their case histories.

I must push back on the suggestion that these real-world complexities pose problems for inferring, in cases where there are closely related nose-colonizing and infecting bacteria, whether the nose-colonizing bacteria are ancestral to the infecting bacteria (e.g. Figure 1), vice versa (e.g. Figure 1), or indeed making no conclusion (e.g. Figure 1). We are able to make these inferences because we leveraged multiple colony sequencing to identify novel mutations that had recently arisen in one population, the hallmark of which being the observation of both ancestral and mutant alleles, and then we observed the new mutant allele fixed in the other population. This same sequencing strategy enabled us to classify mutations that differentiated the nose-colonizing from infecting bacteria (B-class) from those that varied within the nose only (C-class) or infection site only (D-class) with 91% power, as explained in the Materials and methods section Power calculation for within-patient sample sizes.

5) The authors refer to some of the changes that they see in disease isolates as "adaptive" changes, and define adaptive changes in the Introduction as "evolution in response to disease-associated, within-host selection pressures." However, the definition they seem to use throughout the text is more operational: (Introduction) "enrichments in protein altering variants". A clearer definition is warranted. Can the authors clarify if they mean enrichments above baseline or expected rates as might be seen in "relaxed" purifying selection or neutrality, or are they referring more strictly to situations in which d_N_/d_S_ values are significantly greater than 1?

The reviewers are right to insist on a crystal-clear definition of what evidence warrants the use of the term ‘adaptive’ when applied to genetic differences, a term that is the subject of very wide misuse, and frequently applied with little or no more than an assertion.

As the reviewers rightly highlight, we defined this in the Introduction and I have now clarified it to say, “We discovered several groups of genes showing significant enrichments of protein altering variants compared to other genes, indicating adaptive evolution”. This is our principal use of the term, and we describe the precise technical criteria to meet statistical significance in the subsections “Gene enrichment analysis” and “Gene set enrichment analysis”. Occasionally, and where clearly stated, we inferred evidence of adaptation based on the *d_N_/d_S_* ratio, e.g. from a Reference Panel using omegaMap (Wilson and McVean, 2006).

As requested, I have further clarified at the beginning of the Results section “Quorum sensing and cell adhesion proteins exhibit adaptive evolution between nose-colonizing and infecting bacteria” that the gene enrichment analyses detect evidence for adaptation when there is a significant enrichment of protein-altering variants in a particular gene compared to other genes.

The question of whether an enrichment of protein-altering variants compared to other genes is truly indicative of adaptive evolution is an important one. Strong gene-specific and gene-set-specific enrichments of B-class variants are unlikely to reflect relaxed constraint, which should manifest as a fairly homogeneous pattern across genes. Nevertheless, I have introduced a new table (Supplementary file 3) that explicitly presents, for all the significant enrichments shown in Table 3, Neutrality Indices that support the interpretation of these enrichments as evidence for adaptive evolution.

The Neutrality Index (Rand and Kann, 1996, McDonald and Kreitman, 1991) is able to infer adaptive evolution even when a test based solely on *d_N_/d_S_* would not because it does not make the same, overly stringent, null hypothesis of strict neutrality in which equal rates of protein-altering and non-protein altering variation are assumed. Instead, the Neutrality Index effectively compares the *d_N_/d_S_* ratio between populations (i.e. the B-class variants) to the *d_N_/d_S_* ratio within populations (i.e. the C/D-class variants). Selection is expected to have left a stronger imprint on inter-population (B-class) variation because it has had longer to act on older variants. For that reason, a Neutrality Index above one (a greater *d_N_/d_S_* ratio in B- versus C/D-class variants) would be interpreted as adaptive evolution even in cases if the absolute *d_N_/d_S_* ratios were both below one. Note that this test is helpful for interpretation but we did not use it as the main test because it is less powerful than the test for gene or gene set specific enrichments of protein-altering B-class variants.

6) An important distinction in the definition of adaptation might be between adaptation that happened at the colonizing site that was later co-opted in disease, and evolution that truly was "disease-associated". It may be interesting to consider Gould and Vrba's idea of "exaptation" in this context, where changes that evolved in another selective context become beneficial, for another reason, in a new context. It seems possible that the variants that arise in invasive disease may have evolved much earlier under selection in the colonizing site. In this regard, it may be possible to estimate the timing of certain mutations using molecular clock approaches (especially given the multiple isolates collected from each individual). Even if the mutations in question did, in fact, arise very quickly during the process of invasion, it would be interesting to know how much these rates differ from expected rates of evolution. Adding an analysis of inferred rates of evolution would greatly enhance this manuscript.

This is a really interesting point and one that we would have loved to investigate in more detail. I agree that the reviewer’s theory of exaptation is very plausible. For a well-powered statistical analysis, a sampling frame that was equally large but with longitudinal sampling to capture the nose population prior to invasion would be ideal. We have achieved this with one patient (Young et al., 2012), but obtaining large patient numbers currently seems remote.

The BEAST analysis is also a great suggestion, one that we contemplated during the study, and one of several interesting potential follow-up analyses. However, given the current complexity of the paper and the methods used in it, together with reservations over statistical power, we decided against including such an analysis. Nevertheless, the data will be released into the public domain so colleagues interested in pursuing this will be able to as soon as the paper is published.

*7) Toxicity is used throughout the text and is a general term, which can encompass multiple proteins and potential effects. It does not specify the kind of toxicity (e.g. cell death, superantigen activity), target of the toxicity, the effector of the toxicity, or the way it is measured. The general concept of "toxicity" is qualitatively different from phenotypes like "abscess formation" or "quorum sensing" which are specific and measureable. The authors could add clarity using a term like "toxin production", or specifying, "*in vitro *cell death".*

We have clarified by using “toxin production” or “cytotoxicity” where unclear.